# Modeling Carbon-Based Nanomaterials (CNMs) and Derived Composites and Devices

**DOI:** 10.3390/s24237665

**Published:** 2024-11-30

**Authors:** Agustίn Chiminelli, Ivan Radović, Matteo Fasano, Alessandro Fantoni, Manuel Laspalas, Ana Kalinić, Marina Provenzano, Miguel Fernandes

**Affiliations:** 1Materials and Components Division, Technological Institute of Aragon (ITA), María de Luna 7, 50018 Zaragoza, Spain; achiminelli@ita.es (A.C.); mlaspalas@ita.es (M.L.); 2Department of Atomic Physics, Vinča Institute of Nuclear Sciences—National Institute of the Republic of Serbia, University of Belgrade, P.O. Box 522, 11001 Belgrade, Serbia; iradovic@vin.bg.ac.rs (I.R.); ana.kalinic@vin.bg.ac.rs (A.K.); 3Department of Energy “Galileo Ferraris”, Politecnico di Torino, Corso Duca degli Abruzzi 24, 10129 Torino, Italy; matteo.fasano@polito.it (M.F.); marina.provenzano@polito.it (M.P.); 4Department of Electronics, Telecommunication and Computers, Lisbon School of Engineering (ISEL)/Polytechnic University of Lisbon (IPL), Rua Conselheiro Emídio Navarro, nº1, 1959-007 Lisboa, Portugal; mfernandes@deetc.isel.ipl.pt; 5CTS—Centre of Technology and Systems and Associated Lab of Intelligent Systems (LASI), 2829-516 Caparica, Portugal

**Keywords:** carbon, carbon-based materials, graphene, CNT, nanocomposites, electronic, molecular dynamics, continuum models, GFET

## Abstract

A review of different modeling techniques, specifically in the framework of carbon-based nanomaterials (CNMs, including nanoparticles such as graphene and carbon nanotubes—CNTs) and the composites and devices that can be derived from them, is presented. The article emphasizes that the overall performance of these materials depends on mechanisms that operate across different time and spatial scales, requiring tailored approaches based on the material type, size, internal structure/configuration, and the specific properties of interest. Far from attempting to cover the entire spectrum of models, this review examines a wide range of analysis and simulation techniques, highlighting their potential use, some of their weaknesses and strengths, and presenting the latest developments and some application examples. In this way, it is shown how modeling can provide key information for tailoring or designing new materials for specific components or systems or to obtain certain functionalities. At the same time, it is revealed to be an area constantly undergoing development and improvement, as evidenced by the progress made by various of these techniques and the new modeling approaches that have emerged in recent years.

## 1. Introduction

Recent advances in analytical and computational modeling enable prediction and understanding of material properties and responses on scales ranging from the electronic/atomistic, through the microstructure or transitional, and up to the continuum (Figure 1). Multiscale material modeling is based on a systematic reduction in the degrees of freedom at different length scales in which a material can be described. Connections between these scales are established either by parametrization, by grouping (i.e., coarse graining), or by homogenization procedures. These different scales in the description of materials have historically been associated with different disciplines, from physics to engineering, including chemistry and materials science (Figure 1). Each scale can be studied by means of different types of models (electronic, atomistic, mesoscopic, and continuum), which employ distinct basic entities and usually work in different ranges of timescales and number of entities (Figure 2). However, the scale and type of model do not have a one-to-one correspondence, since there are areas of overlap regarding the models that can be employed for a specific analysis.

This article presents a review of different modeling strategies and methodologies used in the field of carbon-based nanoparticles (graphene, carbon nanotubes—CNTs, nanofibers, etc.), carbon-based nanomaterials (CNMs), and devices (such as sensors) that can be obtained from these materials. Within this framework, the variety of models, methods, and applications is extremely wide, and getting a full picture in a condensed report is hard to reach. This review, then, aims to cover some of the key modeling approaches usually employed at different length and time scales, ending with a discussion on a specific type of sensor, derived from these carbon-based materials, as an example of device-level modeling. This leads to the four main sections that constitute the present document.

At the lower scale, the first type of model focuses on electronic calculations and theoretical studies of single-particle and collective excitations. These models are relevant for photonics, plasmonic, and nano-electronics applications. At the next level, molecular dynamics (MD) models enable the study of material properties/responses that are mainly dominated by interactions between atoms or molecules, defined through force fields. These models help in predicting mechanical, thermal, and surface properties, as well as other physical properties such as density, viscosity, or glass transition temperature; they are used transversally in the design of materials and nanocomposites for a wide range of applications. At the continuum level, modeling is typically employed to predict the macroscopic-homogenized properties of materials by considering the constituents as a continuous medium rather than as discrete particles. As in the previous case, these models can be used to predict different types of properties (mechanical, thermal, electrical, acoustic, and other physical properties). Finally, certain kinds of products and devices need to be modeled as systems instead, which might require a combination of different types of properties and physics. Each system has its own requirements, which can lead to the integration of different sub-models. At this scale, as an application example relevant to sensors, this review focuses on the modeling and simulation of graphene field effect transistors (GFET) devices and circuits.

## 2. Theoretical Modeling of Interactions of Charged Particles with Graphene-Based Nanomaterials and Their Composites

Graphene is the best-known and most explored two-dimensional (2D) material. It is the world’s thinnest and strongest material, with the highest electrical and thermal conductivity known. Due to its outstanding physical, chemical, electrical, and optical properties, graphene shows great potential in many fields, including sensors [2,3,4,5,6].

In nanoscale devices, graphene typically appears in stacks separated by insulating layers of finite thickness [7,8,9], which usually support strong Fuchs–Kliewer or optical surface phonon modes [10]. Those phonon modes are active in the terahertz (THz) to mid-infrared (mid-IR) frequency range and can dampen the Dirac plasmon in doped graphene, which operates in the same frequency range [11] or can hybridize with it [12]. As a prototype of layered nanostructures involving doped graphene sheets, a sandwich-like composite that consists of two layers of graphene separated by an insulating layer was studied, and it was found that the structure supports a variety of interesting plasmon-phonon hybrid modes in the THz to mid-IR frequency range [13]. An insulating layer of sapphire (aluminum oxide, Al_2_O_3_) was chosen because it is often used as a dielectric spacer in experiments [14,15,16,17,18,19]. The effective dielectric function of the system was obtained by using a local dielectric function for the bulk Al_2_O_3_ and by using two approaches within the random phase approximation for graphene’s electronic response: an ab initio method based on the time-dependent density functional theory calculations and a method based on the massless Dirac fermion approximation for graphene π bands.

The most efficient way to probe the plasmon-phonon hybridization between graphene and the nearby insulator(s) is by means of an externally moving charged particle. Such hybridization influences the energy loss of an incident particle [20], as well as the resulting wake effect in the induced electrostatic potential [21]. The wake potential (the total potential in the plane of the upper graphene sheet) induced by an external charged particle moving parallel to the graphene-Al_2_O_3_-graphene composite system (Figure 3) was investigated in Ref. [22]. Three velocity regimes of the incident particle were considered in relation to the threshold for excitations of the Dirac plasmon in graphene, given by its Fermi velocity. It was shown that in the low-velocity regime (below the Fermi velocity), only the transverse optical (TO) phonons in the Al_2_O_3_ layer contribute to the wake potential, whereas for intermediate velocities (above the Fermi velocity), the dominant contribution to the wake potential comes from the hybridized Dirac plasmons in the two graphene layers. It was also shown that in the high-velocity regime (well above the Fermi velocity), the dominant contribution to the wake potential originates from the highest-lying hybridized Dirac plasmon. The distance of the charged particle from the top graphene, the thickness of the Al_2_O_3_ layer, and the doping density (i.e., Fermi energy) of graphene were fixed at their respective typical values. The effects of variations of all those parameters were studied in Ref. [23]. It was shown that in the low-velocity regime (below the Fermi velocity), strong effects were observed due to variation of the particle distance, while for the velocity of the charged particle above the Fermi velocity, strong effects were observed due to varying the thickness of the Al_2_O_3_ layer, as well as due to plasmon damping of graphene’s π electrons and graphene doping.

The plasmon-phonon hybridization also has an impact on the stopping force (the dissipative force, which opposes the particle’s motion) and the image force (the perpendicularly oriented conservative force, which bends the particle’s trajectory towards the upper graphene sheet). A thorough analysis of the stopping and image forces on a charged particle moving parallel to the graphene-Al_2_O_3_-graphene composite was performed in Ref. [24], covering broad ranges of the particle speeds and distances, as well as the doping densities of the two graphene sheets. Special attention was paid to the regime of low-particle speeds, where it was found that both forces exhibit an interesting interplay between the hybrid modes of the phononic type and the continuum of the intraband electron–hole excitations in graphene and where the modeling was sensitive to the temperature effects and the treatment of the phenomenological damping. It should be noted that the stopping force is the negative of the usual stopping power, whereas the image force is related to the familiar image potential.

Note that the wake effect and the stopping and image forces were previously investigated theoretically in free-standing and supported graphene [20,21,25,26,27,28,29,30,31,32,33,34,35,36,37,38,39,40,41,42,43,44].

In the last few years, van der Waals (vdW) heterostructures based on graphene and hexagonal boron nitride (hBN) layers with different stacking modes have attracted a great deal of interest because of their potential applications [45,46,47,48,49,50,51,52,53]. Graphene/hBN vdW heterostructures were studied very recently in Refs. [54,55]. In Ref. [54], the authors theoretically examined the 2D plasmon excitation efficiency in graphene/hBN composites via silver spherical nanoparticle, whereas in Ref. [55], the authors theoretically investigated the Bloch plasmon polaritons band structure in Al_2_O_3_/graphene/hBN/graphene nanoribbons assembly depending on the doping of graphene layers.

Electron energy loss spectroscopy (EELS) is a commonly used experimental technique for investigating electronic and plasmonic properties of 2D materials and vdW heterostructures [56,57,58]. Theoretical modeling of the experimental EELS data for free-standing (single- and multilayer) graphene sheets obtained by (scanning) transmission electron microscope ((S)TEM) is presented in Refs. [59,60,61,62], whereas the theoretical modeling of the experimental EELS data for monolayer graphene supported by different substrates is given in Refs. [63,64,65,66]. In Refs. [59,61], the authors treated the multilayer graphene as a layered electron gas with the single-layer polarizability modeled by a 2D, two-fluid hydrodynamic (HD) model [59] and with ab initio calculations [61] and found good agreement with experimental EEL spectra. In Ref. [60], the authors presented momentum-resolved EEL measurements of free-standing, single-layer graphene and compared them with corresponding time-dependent density functional theory calculations. Good agreement with experiments was found for finite momentum transfers if crystal local-field effects were considered. An analytical modeling of the experimental EELS data for free-standing graphene obtained by STEM is presented in Ref. [62]. The authors used an optical approximation based on the conductivity of graphene given in the local, that is, frequency-dependent form derived by both a 2D, two-fluid extended HD model and an ab initio method and found very good agreement with the EEL spectra from three independent experiments. In Refs. [63,64,65,66], the authors used the 2D, two-fluid HD model for interband transitions of graphene’s π and σ electrons and an empirical Drude–Lorentz model in the local approximation for metal substrates to reproduce the experimental EELS data. A very good agreement was obtained for monolayer graphene supported by Pt(111) substrate [63,64] and Ru(0001) substrate [64] for high-quality graphene grown on peeled-off epitaxial Cu(111) foils [65] and for graphene contacted with a Pt skin [66]. The model failed for the case of monolayer graphene supported by the Ni(111) substrate [64], presumably due to the strong hybridization between the π states of graphene and the *d* bands of Ni, which was not accounted for in the model.

Acoustic plasmon (AP) in graphene or in graphene-dielectric-metal structures has been studied very intensively in the last few years [67,68]. In Ref. [69], the authors focused on the AP in graphene doped by alkali metals and demonstrated that two isoelectronic systems, KC_8_ and CsC_8_, support substantially different plasmonic spectra, that is, the KC_8_ supports a sharp Dirac plasmon (DP) and a well-defined AP, while the CsC_8_ supports a broad DP and does not support an AP at all. These findings could be very useful in the area of chemical or biological sensing [70,71].

## 3. Molecular Dynamics Applied to CNM Properties Prediction

Molecular dynamics (MD) is a computational method to simulate the behavior of atoms and molecules over time. In an MD simulation, the trajectories of atoms (considered rigid spheres) and molecules are determined by numerically solving Newton’s equations of motion. Forces between particles are calculated by exploiting calibrated interaction potentials (force fields), and from these forces, the accelerations, velocities, and subsequent positions of the atoms are determined [72]. For materials science, MD provides detailed insights into phenomena that would otherwise be difficult to observe directly, such as diffusion, phase transitions, interfacial effects, and mechanical behavior at the nanoscale.

One of the key aspects of MD simulations lies in the force fields, which are mathematical models used to calculate interactions between atoms due to the presence of covalent bonds (bonded interactions) and electrostatic forces (Coulomb/van der Waals and non-bonded interactions) [73]. In MD simulations of CNMs, some specific force fields have gained prominence [74]:The Adaptive Intermolecular Reactive Bond Order (AIREBO) potential is tailored for carbon systems and describes long-range van der Waals interactions and torsional effects. It is versatile for modeling both sp^2^- and sp^3^-hybridized carbon structures [75]. AIREBO might not perform well for systems with significant charge transfer or in the case of interactions with elements outside its parameterization.The Tersoff potential considers both the distance between atoms (bond lengths) and their relative orientation (bond angles) to provide a detailed representation of the complex interactions that occur in carbon-based materials [76]. This potential may not be ideal for modeling weak interactions, and it might require recalibration for systems different from its original parameterization.ReaxFF is a reactive force field capable of simulating bond formation and breaking during MD simulations. This dynamic nature is achieved by not predefining specific bond types but allowing the system to evolve based on atomic positions and interactions. Due to its reactive nature, ReaxFF can be computationally demanding. It also requires careful system-specific parameterization to ensure reliable results, for example, in the case of condensed carbon phases [77].Machine learning (ML) interatomic potentials differ from traditional ones, as they do not depend on fixed mathematical formulas. Instead, they learn representations of the potential energy surface of the system through trainings based on lower-scale simulations. Several implementations for certain carbon forms with near DFT-level accuracy have been reported in the literature, for example, Gaussian Approximation Potential (GAP) [78], hybrid neural network potential [79], GAP-20 potential for various crystalline phases of carbon, and amorphous carbon [80]. Furthermore, MACE—a transferable force field for organic molecules created using ML trained on first-principles reference data—was recently implemented [81]. Despite the good accuracy of current ML-based force fields in predicting the properties of carbon allotropes, various challenges still exist, especially regarding the description of mechanical properties and the curation of reliable training datasets.

MD is well suited for investigating various properties of CNMs and composites made thereof [82]. The mechanical properties of these materials, for example, can be determined through MD simulations that allow the stresses and strains experienced by the system to be evaluated [83]. In these tests, a strain is systematically applied to the system, and the resulting stress responses of the material are recorded, thus providing information on its elastic constants, tensile strength, and potential fracture points (Figure 4A) [84]. For instance, simulations of cellulose nanocrystal-graphene composites revealed enhanced mechanical properties due to covalent bonding and van der Waals interactions [85]. Similarly, MD analyses of single-walled carbon nanotubes (SWCNTs) have demonstrated Young’s moduli in agreement with experimental values, showcasing their exceptional mechanical strength and stiffness [86]. However, these simulations face some challenges, as the simulated strain rates should be much higher than those typically found in experimental setups to provide meaningful results in a feasible computational time. Another technique for analyzing the mechanical properties of a material is nanoindentation, which involves the simulation of a virtual indenter pressing on the surface and allows the hardness and localized stress response to be derived [87]. For example, nanoindentation MD simulations on polymer nanocomposites highlighted the effect of nanoparticle interactions and temperature on mechanical reinforcement [88]. The results, though, can be influenced by the chosen shape of the indenter and the interaction potentials utilized. Pull-out tests simulations [89], and density profile analysis [90], however, are valuable aids in characterizing the interface region between a CNM and the surrounding environment (e.g., the polymer in the case of composites, see Figure 4B), which is a key interaction region that strongly determines the properties of the composite material. These analyses provide useful insights into the mechanisms of load transfer through materials and interfacial adhesion behavior.

The thermal properties of CNMs can be explored using various protocols [91,92]. In the Non-equilibrium Molecular Dynamics (NEMD) technique, for instance, a temperature gradient is established within the simulation domain, thus enabling the calculation of thermal conductivity [93]. NEMD was employed to calculate the thermal conductivity of multi-walled CNTs with different geometrical features, such as diameter, length, chirality, and number of walls (see Figure 4C) [94]. Still, this method entails an issue, as the artificial imposition of a gradient might not realistically replicate actual experimental scenarios. The Equilibrium Molecular Dynamics (EMD) method offers another approach, relying on the analysis of heat current fluctuations within a system at equilibrium. A notable example involves the use of EMD simulations to determine the thermal conductivity of graphene nanoribbons [95]. By employing the Green–Kubo method, researchers studied the effects of ribbon width, edge roughness, and hydrogen termination on thermal conductivity. The results showed that smooth edges yield the highest conductivity, while edge roughness significantly reduces it. However, EMD often requires extended simulation times. Another critical thermal property that can be computed by MD simulations is the thermal boundary resistance (TBR, also known as Kapitza resistance, originating from phonon scattering in the presence of defects or interfaces) [96]. As carbon nanomaterials are often embedded into other materials (e.g., polymeric matrices and fluids, see Figure 4D), understanding the efficiency of heat transfer across these interfaces becomes vital [97]. Both NEMD and EMD protocols offer quantitative insights into this property. Furthermore, theoretical models such as the Acoustic Mismatch Model and the Diffuse Mismatch Model, informed by inputs from MD simulations, provide complementary perspectives on TBR [98].

For thermodynamic properties, MD simulations lean on specialized techniques, like free energy calculations. Properties such as adhesion can be probed using advanced sampling methods, such as umbrella sampling and metadynamics [90], but these techniques often require care in choosing the force field and can be computationally burdensome. The wetting properties of CNMs are fundamental as well for optimizing their performance in suspensions and composites. Wettability, quantified by contact angle measurements, can indeed affect the dispersion of CNMs in various solvents, along with the ability of polymers to spread and adhere to CNMs, thus influencing the mechanical and thermal properties of the composite [99]. At the atomistic level, two main approaches are used to measure the contact angle. The free energy perturbation (FEP) method involves calculating the free energy changes to determine the interaction parameters between a liquid and a surface [100]. This approach allows the interaction parameters to be calibrated and the work of adhesion and friction coefficients to be evaluated. In the second method, called the droplet method, a liquid droplet is placed on the tested surface and allowed to relax until equilibrium is reached [101]. The contact angle is then measured by analyzing the shape of the droplet along the three-phase contact line [102]. The latter approach is widely used to study the effects of surface roughness and interfacial properties. For instance, MD simulations revealed that difunctional epoxy and cyanate ester resins exhibit high wettability on CNT surfaces, while polyether ether ketone resins show poor wetting properties [103]. Graphene oxide, due to its hydrophilic functional groups, generally offers better wettability than pristine graphene, improving dispersion and interfacial bonding in polymer matrices [104]. Additionally, carbon nanoparticle coatings synthesized through controlled flame deposition can be tailored from hydrophilic to superhydrophobic states by adjusting synthesis conditions, which optimize surface interactions and enhance composite performance [105].

Other types of properties that can also be studied through MD models are the dielectric ones (dielectric constants, relaxion) using specific approaches such as the dipole moment fluctuation method [106].

While offering mechanistic insights into the properties and behavior of CNMs, MD simulations do not lack challenges. For instance, MD approaches typically operate within specific temporal (fs to µs) and spatial (nm to µm) scales, and phenomena outside these scales might not be detected and analyzed effectively [107]. Moreover, the choice of the force field can significantly impact the results, as not all force fields describe the intricate interactions in nanomaterials with the same accuracy [108]. Furthermore, high-resolution simulations, especially those involving long timescales or large systems, can be computationally demanding, requiring substantial resources and time. The imposition of periodic boundary conditions, instead, can lead to artifacts in the results, especially if the size of the simulated system is not large enough compared to the phenomena under observation [109]. Finally, the rates used in simulations (e.g., thermal or mechanical ones), due to computational constraints, often exceed experimental rates, potentially leading to discrepancies between numerical and experimental data [110].

Looking ahead, the MD research landscape on CNMs presents promising prospects [111]. One of the most significant improvements in recent years has been the development of enhanced force fields. Ongoing research in this area aims to refine these force fields specifically for carbon nanomaterials, with the goal of increasing the accuracy in predicting their properties. Moreover, the growing popularity of multiscale modeling as a robust approach offers significant opportunities: by linking/coupling MD with other simulation methodologies, such as electronic, mesoscopic, and/or continuum models [112], researchers aim to bridge the spatiotemporal gaps between scales. Yet, one of the most transformative shifts in the field of molecular dynamics is probably the integration of data-driven approaches within MD simulations. The support of machine learning and artificial intelligence is not merely augmenting computational efficiency, but it is reshaping the paradigms of simulations. These tools offer optimized parameter selections, predictive capabilities, and the prospect of devising new force fields. Lastly, shared efforts between experimentalists and computational researchers are fostering iterative refinements in simulation methodologies. This synergy is guiding MD studies closer to experimental observations, ensuring a more harmonized understanding of CNMs.

**Figure 4 sensors-24-07665-f004:**
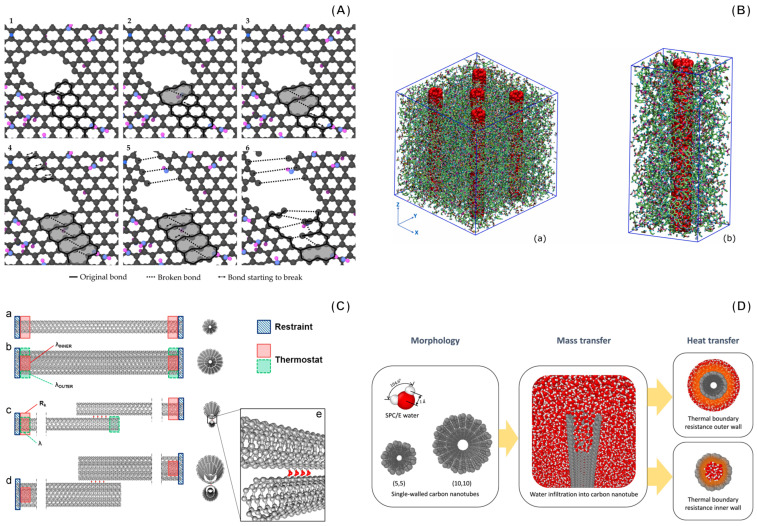
Examples of atomistic models used to study carbon structures. (**A**) Dislocation formation during stretching simulation of a graphene oxide nanoribbon model: snapshots of the simulated trajectory, with time evolution from 1 to 6. Image from C. Saenz Ezquerro et al., CC BY 4.0 [83]. (**B**) Epoxy resin boxes reinforced by (**a**) 5 single-walled carbon nanotubes (SWCNTs) and (**b**) one SWCNT. Image from S. Mohammad Nejat et al., CC BY 4.0 [91]. (**C**) Investigation of the thermal properties of (**a**) SWCNTs and (**b**) double-walled carbon nanotubes (DWCNTs) composed of two coaxial SWCNTs. Computation of thermal conductivity and Kapitza resistance for a network of (**c**) SWCNTs or (**d**) DWCNTs joined by (**e**) C-O-C bonds. Reprinted from [93], Copyright 2015, with permission from Elsevier. (**D**) Effect of surface wettability on heat and mass transfer at the interface between water and CNTs. Image from A. Casto et al., CC BY 4.0 [99].

## 4. Continuum Models

At the next level within the multiscale materials modeling framework, continuum models are powerful tools for predicting the properties of nanocomposites. These models consider the microstructure of the nanocomposite, including the size, shape, and distribution of the nanoparticles, to predict the macroscopic-homogenized properties of the material [113,114,115]. Compared to the previous ones, continuum models can handle much larger systems but lack the detailed description of atomic or molecular interactions. As mentioned, they can be used to predict different types of properties or responses, such as mechanical, thermal, electrical, and acoustic, among other physical properties [116].

There are basically three types of continuum models, namely, analytical, semi-analytical, and numerical [117]. The most common analytic models used for nanocomposites are based on the early study by Eshelby [118] on ellipsoidal inclusions in an infinite elastic matrix, such as the relevant Mori-Tanaka mean field approach [119], which introduced as a hypothesis that each particle sees as far field the average matrix strain, allowed the application to non-diluted concentrations, calculating the properties of the composite by a physical combination of the properties of the nanoparticles and the matrix, weighted by their respective volume fractions. Other types of analytical models are the self-consistent models (SCMs), where each nanoparticle in the nanocomposite is embedded in a matrix that has the same properties as the nanocomposite itself (this leads to a system of coupled equations that must be solved numerically). In general, these models can be solved relatively quickly compared with other approaches, making them useful for initial approximations and quick calculations. Despite this, as it is reported later, more and more advanced models are being developed attempting to capture complex geometric and material aspects. There are hundreds of studies in the literature applying these models to determine standard fiber-based composite properties first and later applied also to nanocomposites, for example, considering the matrix-particle interphase as a third material phase constituent. In this sense, interesting references comparing/reviewing all these methods are the article by Y. Wang and Z. Huang [120], the chapter of Yehia A. Bahei-El-Din about Averaging Models of Fibrous Composites in Ref. [121], and the recent publication of A. Elmasry that presents a review comparing different models for effective properties calculation of nano- and micro-composites [122]. Studies focused on mechanical properties include the following:(i)Nanocomposites’ stiffness prediction (i.e., response in the elastic range), as the research performed by M.M. Shokrieh for an epoxy resin modified with graphene nanoplatelets [123], the study presented by A. Chiminelli and M. Laspalas for an epoxy resin modified with MWCNTs (see Figure 5A) [124], the study by A. Singh and D. Kumar studying the influence of the functionalization of graphene nanoplatelets in the elastic properties of a modified polyethylene [125], or a more recent study by D. Shin for graphene-modified PET [126].(ii)Non-linear behavior and strength predictions of CNMs, as the approach proposed by J. Nafar Dastgerdi introducing interfacial damage/debonding processes in CNTs-reinforced polymers [127] or the model developed by W. Azoti and A. Elmarakbi applied to a graphene platelets GPL-reinforced polymer PA6 composite (Generalized Mori-Tanaka) [128].

Other studies can be found about the utilization of this type of analytic approach to develop more advanced models introducing viscoelasticity (see Figure 5B) [129] or creep effects [130], among others. Overall, good correlations with experimental results are obtained for the elastic properties, especially at low nanoparticle concentrations, while at higher concentrations, more significant deviations are usually observed. This is generally explained due to interactions between the nanoparticles at high contents that produce non-homogeneous dispersions and agglomerations. For predictions in the non-linear regime and in terms of strength, it is generally seen that the results are strongly dependent on the particle/polymer interface representation and on the aspect ratios of the particles. This is also observed in the modeling of other types of properties of nanocomposites (thermal, electrical, acoustic, etc.). For example, in the study published by J. Shao [131] for predicting dielectric properties of polymers modified with nanoparticles, the Knott model was modified with the Tanaka formula for hybrid particles to take into consideration the interphase region. Obviously, the issues mentioned at high concentrations are also present in these cases. Apart from this, these models are valuable to evaluate different functionalizations of carbon-based nanoparticles.

In addition to the models mentioned above, some semi-empirical analytical models are also well-known for composites, starting with the classical rule of mixtures, evolving to Chamis’ model and the Halpin–Tsai’s model (as a simplified version of the SCM), and leading to more advanced models introducing (again) elastoplasticity (approaches to yield stress and linearization methods [120]). The Halpin–Tsai approach for aligned reinforcement has been employed widely for the analysis of graphene-based nanocomposites [132]. In this sense, studies such as the ones developed by Weon [133], Chong [134], and Zarasvand and Golestanian [135] can be highlighted (see Figure 5C). In the last study, the non-linear tensile stress–strain behavior of randomly distributed graphene nanocomposites has been obtained, presenting a good correlation with experimental results in the whole range of the stress–strain curves. In a study published by M. Yang [136], the Halpin-Tsai model was adapted to quantitatively characterize the effect of temperature on the yield strength of nanofiller-reinforced polymer–matrix nanocomposites. Compared with the classical ones, the model showed better agreement with the available experimental data from subzero temperature to the full glass transition region. Progressing with elastoplasticity, several Non-linear Mean Field Methods have been developed from the elastic ones, the Tangent and Secant approaches being the most known. Some reference publications implementing these methods are the ones of Doghri [137,138]. An advantage of these models is their straightforward implementation. They also allow the study of more complex loading cases than other models, including cyclic loads.

Semi-analytical methods are based on global constitutive equations that are evaluated from the local scale using analytic/explicit relations that link the microscopic and the macroscopic properties. Analytical relations are usually dependent on mean field procedures. The best known is transformation field analysis (TFA). It connects analytical and computational approaches by a computational evaluation of the localization operators. The main concept is replacing the plastic strain field with piecewise uniform fields to reduce the number of macroscopic internal variables. Thereby, a set of reduced constitutive relations for the heterogeneous material can be established, leading to a computational time reduction compared to full numerical models [126]. Reference studies of this type of model are the ones published by Dvorak [139,140]. A more recent study published by I.A. Khattab and M. Sinapius presents an interesting implementation of the TFA model as a user routine integrated into RVE-micro-scale model [141]. The results reveal that the TFA is a proper method for solving inelastic deformation and other incremental problems in heterogeneous media with many interacting inhomogeneities on a nanoscale level.

Finally, numerical continuum micromechanics models, such as the finite element method (FEM), are also considered powerful approaches for CNM modeling. In numerical approaches, an acknowledged constitutive model (elasticity, viscoelasticity, elastoplasticity, and viscoplasticity) is assumed in a Representative Volume Element (RVE) to induce an explicit macroscopic model. On one hand, these models allow for a more accurate description of the materials nanomorphology [142]. This is one of the limitations of analytic models, which usually requires an idealization of the nanoreinforcements’ shape, representing them as discs, cylinders, or spheres. On the other hand, these models allow for the introduction of complex, non-linear, multi-phase material behaviors. The main disadvantages of finite elements are computational costs, size dependency of the results, and limitations in the development of RVEs with high inclusion volume concentrations and aspect ratios.

RVE-based models apply to statistically homogeneous materials. They can be used as a repeating unit cell (RUC) when the (nano)composite has a periodic microstructure. A sufficient number of randomly distributed fibers or particles to be contained in the RVE is needed so that the microstructure of a composite could be reflected precisely [120]. This is also linked with the size of the volume studied. Various studies reflect the importance of defining a proper RVE size [143]. In this sense, it happens that a larger RVE size gave better prediction accuracy but resulted in lower computational efficiency. Finally, the boundary conditions are a critical aspect of RUC models, and they have to be defined carefully to properly represent the effect of the reinforcements/particles distribution patterns and the loads applied.

Some first reference publications that can be found about RVE and FEM to estimate properties of polymer nanocomposites are the studies of Liu and Chen [144,145]. Particularly, they studied the elastic properties of CNT-reinforced polymers. Chwał and Muc [146,147] applied a similar approach with various boundary conditions to calculate the mechanical properties of SWCNT-polymer nanocomposites. More recent studies present a combination of this type of FE continuum model with MD, in a similar way to the multiscale framework presented in this review. For example, recently Barakat et al. published an article where the distribution of mechanical properties of graphene-based polymer nanocomposites is computed using a micro-meso to macro hierarchical computational approach employing non-equilibrium atomistic MD simulations and continuum FE models [148]. In the same line, the study published by Muhammad et al. can be highlighted [149], where not only mechanical but also thermal properties are predicted for graphene-reinforced epoxies. For SWCNT-polymer nanocomposites, Malague proposed a procedure to assess size effects using the atomistic simulations and equivalent continuum model with a large number of CNTs [150].

Elastic RVE/RUC models can be extended to non-linear cases, providing that the non-linear constitutive laws for the constituents are available. For example, based on a three-dimensional RVE model, Yuan and Lu [151] conducted a numerical investigation on the elastoplastic behavior of carbon nanotubes (CNTs) reinforced polymer composites (see Figure 5D). In Ref. [152], Zarasvand also conducted experimental, numerical, and micromechanical studies to determine the non-linear behavior of CNT-reinforced polymer. When non-linear constitutive behaviors are considered, standard FEM approaches become computationally consuming. To address this limitation, several numerical strategies with reduced computational effort have been developed. Examples are the Voronoi Cell Finite Element Method (VCFEM) [153,154], the Generalized Method of Cells (GMCs) [155], and the Finite Volume Direct Averaging Micromechanics (FVDAM) [156,157]. However, sometimes even in these cases, the computation times are excessive [110,122], so it is expected that in the coming years we will continue to see the development of new continuum modeling approaches and methodologies.

**Figure 5 sensors-24-07665-f005:**
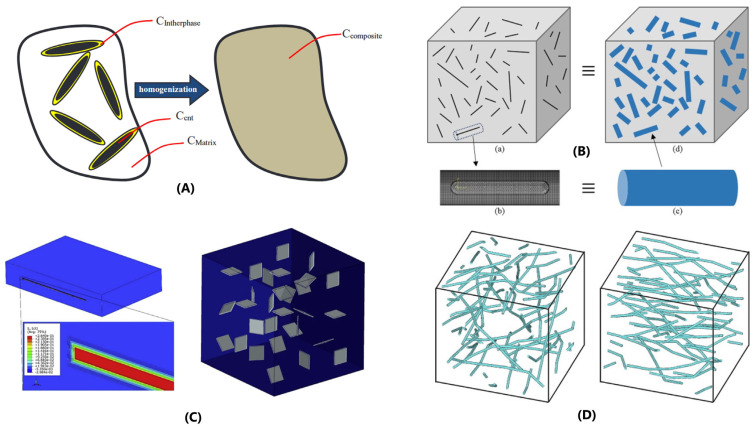
Examples of continuum models to study CNMs: (**A**) mean field homogenization scheme applied to an epoxy resin modified with MWCNTs [124], (**B**) combined FE and analytic Mori-Tanaka models for CNT-reinforced polymer taking into account viscoelastic properties as example of a hybrid approach [129], (**C**) utilization of Halpin–Tsai approach for Graphene Nanoplatelet reinforced epoxy to determine the non-linear stress–strain behavior [135], and (**D**) REV/FEM model analyses considering elastoplasticity for wavy and random CNT distribution in nanocomposites as example of a numerical continuum micromechanics model [151].

Finally, at this scale, the integration of continuum models with data-driven ones and ML techniques is also an active area of research and development. An overview of the application of these approaches with different purposes is given in the article of F. Bock et al. [158]. In Ref. [159], a full-scale stochastic analysis from nano-, micro-, meso- to macroscales is presented for thermal conductivity prediction in CNT polymeric nano-composites. In this study, it is shown how ML can help to significantly reduce modeling computation costs. With this data-driven enhanced multiscale strategy, an uncertainty quantification was obtained at different length scales for different properties, allowing to determine the reliability of the predictions. Another example is the methodology proposed in Ref. [160], based on sequentially coupled electro-mechanical FE simulations (micro-scale FE models) combined with an artificial neural network able to predict the electrical conductivity tensor as a function of the applied strain. This approach enables simulations at the macro-scale, which cannot be performed by explicit modeling of the microstructure due to the computational cost. Significant progress is foreseen in the development and application of these hybrid physical-based and data-driven modeling methodologies. While this integration of models and techniques is promising, challenges remain, such as the need for large, high-quality datasets and the computational cost of training complex models.

Based on what has been described, future research will likely focus on improving data acquisition methods and developing more efficient algorithms. In addition, more comprehensive and standardized validation frameworks are needed to ensure the reliability and broader application of these modeling techniques in CNMs. This should result in greater implementation of these modeling techniques in commercial or open-source software.

## 5. CNM Devices—Graphene Field Effect Transistors

When we arrive to discuss carbon nanomaterial devices that can be considered mature enough to be used in sensing systems, one of the most used configurations is based upon graphene field effect transistors (G-FET). High-frequency applications have been, for the first time, a major goal for this class of devices, and a series of interesting results reporting the fabrication of research-level devices working in the GHz domain were published in 2010 [161,162,163,164]. Anyway, the problems that were initially addressed for the graphene transistors to replace silicon MOSFET as the choice alternative for commercial high-performance logic and high-frequency electronics [165] prevented until now their large-scale adoption for this class of applications.

As a follow-up, the intense research study realized during these years has produced many interesting results, experimental and theoretical as well, that have permitted a good insight into the physic mechanism of the GFET structures, paving the way to the definition of an application playground in the biosensing domain, avoiding the HF requirements of a G-FET-based ASIC project. A nice summary of the present state of this approach can be found in the recently published review paper by S. Szunerits [166]. According to the authors, not only graphene is proposed for FET biosensing applications, but also other 2D materials like metal dichalcogenides, hexagonal boron nitride, or black phosphorus have been investigated for FET gate channel technology. Indeed, graphene maintains its position as a first-choice material for commercial applications for low-term potential.

Although it is not currently available on the market, it may be foreseen that the use of GFET circuitry will soon find commercial application in wearable flexible sensing systems [167], in point-of-care systems [168,169], and even in skin bioelectronics for health monitoring [170], exploiting its mechanical/electrical properties together with biocompatibility characteristics. Advanced experimental results in this field have been recently presented for applications of major interest, like, for example, cancer monitoring [171], COVID-19 screening [172], or glucose monitoring in diabetic patients [173].

While no clinical validation has yet been performed, the proposed technology for these important biomedical playgrounds has been fully demonstrated in laboratory environments. The next step still needed for allowing the fabrication of large-scale point-of-care biosensing systems is the definition of a reproducible fabrication process, integrated with standard silicon CMOS technology. Once some variability can be expected from the fabrication condition, modeling and simulation will play a major role in the G-FET circuitry design. Looking at the recent past, modeling and simulation of GFET devices and circuits in a variety of configurations has certainly made an important contribution to the actual advanced state of the art for design and fabrication.

Regarding the electrical properties, a model to calculate the DC characteristics of large-area graphene field-effect transistors was initially presented by Thiele and Schwierz [174]. This paper opened the door to the heuristic approach of correlating the experimental results with the specific characteristics of the graphene material, like the sheet carrier density and transport properties. More specifically, the Thiele and Schwierz model successfully describes the experimental DC behavior of the drain current (Id) as a function of the Drain-Source Voltage (Vds) for different Gate-Source voltages (Vgs), using a reduced set of parameters about the device geometry like the gate length (L) and width (W), the insulator thickness (tox), and drain and source contact resistances (Rs, Rd), together with material properties like charge mobilities in the graphene channel, sheet carrier density, and carrier saturation velocity.

As a general description, a G-FET has a structure that can be thought of as similar to a thin film transistor (TFT), where the 2D graphene layer is used for channel forming. The traditional approach for the electrical simulation of semiconductor FET devices is based on the drift-diffusion model, including the Poisson equation [175]. Following an approach similar to the one proposed by Thiele and Schwierz, yet including the standard drift-diffusion approach, a Verilog compact model suitable for circuit-level design based on the drift and diffusion scheme has been presented and made freely available by Landauer [176]. The main problem for the direct application of this model to the G-FET structure is the definition of the charge mobility value, which in a 2D material can be largely affected by the presence of impurities, lattice defects, and substrate quality. A nice approach to this problem was recently described by Nastasi [177]. They describe extensively the physical and numerical model, including the model for the mobility and its dependence on the applied electric field, and the results obtained on a top gate configuration described by a 2D geometry.

The approach directed to modeling the behavior of the device when embedded in a more general circuit is of major importance when the target application is related to the sensing of some specific bioelement. From this point of view, the correlation between material properties and electrical behavior and output extraction is the main objective of any modeling task. The G-FET structure better suited for targeting the sensing of biomolecules (such as proteins and nucleic acids) in biological fluids is the co-planar and liquid-immersed-gate configuration, which is effectively largely preferred in biological G-FET design. This configuration overcomes the standard top-gate or bottom-gate inherited by the flat panel semiconductor industry and presents a new challenge for modeling and simulation approaches. The main results of this activity are the production of G-FET models suitable to be used in a circuit simulator to support the fabrication of complete G-FET-based circuits for biosensing applications. A dual-gate G-FET model for circuit simulation, suitable to be used for biosensing use, was presented by Umoh [178]. The main value of this model is the direct application in a SPICE implementation, in a traditional configuration settling on standard FET SPICE model parameters, used internally to calculate junction resistances and capacitances. Within this context, Jmai [179] presented a model freely distributed in its legacy MATLAB and Verilog (version IEEE 1364.2001) implementation, allowing the user to select an appropriate topology for a system-level design suitable to be used in real-life applications. Once again, the charge mobilities are a main parameter that needs to be assessed as a function of the fabrication condition. As already mentioned, considering the multiscale concept, these models at system level can be fed with results obtained from models at lower scales, in the specific case of GFETs, for example, to determine the dielectric properties that are required. In this way, the nanocomposite properties can be investigated and designed with MD or continuum models, and these results can be introduced in the transistor model for device optimization.

As a matter of fact, from the biosensing point of view, it is possible to assume that the presence of the target bioelement on the graphene surface would be the main driving cause for the mobility fluctuation and the correspondent variation of the drain current of the transistor. This observation focuses the modeling attention on a differential current extraction, targeting the information about the presence of the biomarkers instead of the mobility value itself. The model for the dual-gate configuration is directly extendible to the liquid gate configuration and a graphical comparison between the two structures can be seen in Figure 6. From a higher application-level approach, Fuente-Zapico has adapted the Synopsys Sentaurus Commercial TCAD suite for the simulation of a liquid gate graphene field effect transistor, GFET, used as an antibody-based biosensor [180]. In this study, the authors have successfully included the liquid electrolyte, the antibody functionalization of the graphene surface, and the biomarker trapping effects, reporting application-driven parameters to the current–voltage characteristics calculated in a standard commercial device circuit simulator.

As a bottom line, it must be outlined that The Graphene Flagship’s 2D Experimental Pilot Line (2D-EPL) has been offering, starting in 2022, prototyping services to academics, SMEs, and companies, which can benefit from the progress of graphene-related materials integration with silicon [181]. The 2D-EPL provides a multi-project wafer run (MPW) for the G-FET circuit, including a top/bottom contact with an optional local or global back and liquid gate. Directed to bio/gas/chemical sensors, this MPW includes a stand-alone G-FET circuit as well as a full CMOS integration.

## 6. Conclusions

A review of different modeling strategies and methodologies used within the framework of carbon-based nanoparticles (graphene, carbon nanotubes—CNTs, nanofibers, etc.), carbon-based nanomaterials (CNMs), and devices (as sensors) that can be obtained from these materials has been presented. The range of modeling and simulation techniques generally employed covers electronic to continuum models, passing through atomistic and meso-scale. For each case, their typical uses, some of their weaknesses and strengths, and the latest developments are presented, including application examples and key references. Sometimes, multiscale workflows are required, where these different models and scales are sequentially linked as part of a single analysis. Modeling devices require multi-physics models, integrating different phenomena that can be present simultaneously. Then, system-type models are used. In this sense, the modeling of G-FET for sensing applications is included in this review as a specific example of carbon-based devices.

Regarding the theoretical study of interactions of charged particles with graphene-based nanomaterials, a mathematical framework for studying such interactions uses various response functions of such materials as an input, which may be available from analytical phenomenological models, ab initio calculations, or even experimental data from optical spectroscopies. This allows various research groups to quickly develop reliable models for analyzing plasmonic properties of layered nanostructures, including graphene, which can be used to predict the properties for novel designs of such structures.

Regarding molecular dynamics models, they are typically used to investigate various properties of CNMs at the atomistic scale, thus helping to predict their physicochemical behavior and optimize their performance for specific applications. MD simulations can provide valuable insights into the thermomechanical properties of carbon-based structures (e.g., thermal conductivity or elastic constants) and interface interactions in composite materials, yielding increasingly accurate results due to ongoing research efforts and the integration of data-driven approaches.

Continuum models include mainly analytical, semi-analytical, and numerical approaches. Analytical models can be solved relatively quickly and often provide a clear and straightforward understanding of the underlying physics and insights into the relationships between different variables and parameters. However, sometimes they are limited in terms of the geometrical complexity that they can address. Numerical models can partially deal with these issues but are constrained by their computational cost, size dependency of the results, and restrictions in the development of RVEs with high inclusion volume concentrations and aspect ratios. Overall, good reproducibility of the properties of the materials is obtained, especially at low nanoparticle concentrations, even for non-linear and time-dependent responses, while at higher concentrations more significant deviations are usually observed. This is generally explained due to interactions between the nanoparticles at high contents that produce non-homogeneous dispersions and agglomerations. For mechanical property predictions in the non-linear regime or in terms of strength, it is generally seen that the results are strongly dependent on the particle/polymer interface representation and on the aspect ratios of the particles. Similar influences are also observed for other types of properties, such as thermal ones. Apart from this, these models are also being hybridized with data-driven ones and ML techniques, looking for more efficient strategies that can allow them to run a high amount of relatively complex simulations.

Finally, the electrical behavior of G-FETs, their analysis, and their design can be supported through Spice and Verilog models made freely available by the authors. The theoretical formulation of such models is available in the literature, and the main simulation software houses are starting to include them in their commercial packages. We can expect, for the near future, the introduction of integrated models, describing mechanical, thermal, electrical, and optical models, addressing the sensing applications where this class of devices can offer breakthrough solutions. Modeling techniques are powerful tools to support materials research in the development of novel applications, particularly in the case of carbon-based materials and their derived composites and devices. It provides the key information for identifying new materials, tailoring materials, and/or designing materials for structures and systems.

Considering the huge potential for a short-term commercial evolution in the field of low-cost sensing devices based on CNM material and composites, simulation models are expected to accompany this trend and evolve in complexity and reliability. Along this review, a variety of different approaches to modeling the physical properties of CNMs have been highlighted, and from the point of view of the sensing applications, it will be important to produce a robust description about how these physical properties may interface to the real world. While in the academic world there is clear and productive activity in model optimization across multiple scales—Table 1 reports a selection of successfully linked multiscale models of CNM-based composites taken from the recent scientific literature—presently, we are still missing the intervention of the big players from the market of the commercial simulation packages that should be able to define a standardization path like the one we have witnessed in the recent past for ASICs of FPGA platforms. Standardization is the keyword for the next step in the CNMs model evolution, allowing the integration with state-of-the-art microelectronic design and technologies; it will represent the boarding pass for the market penetration in the IoT/SmartCities playground.

## Figures and Tables

**Figure 1 sensors-24-07665-f001:**
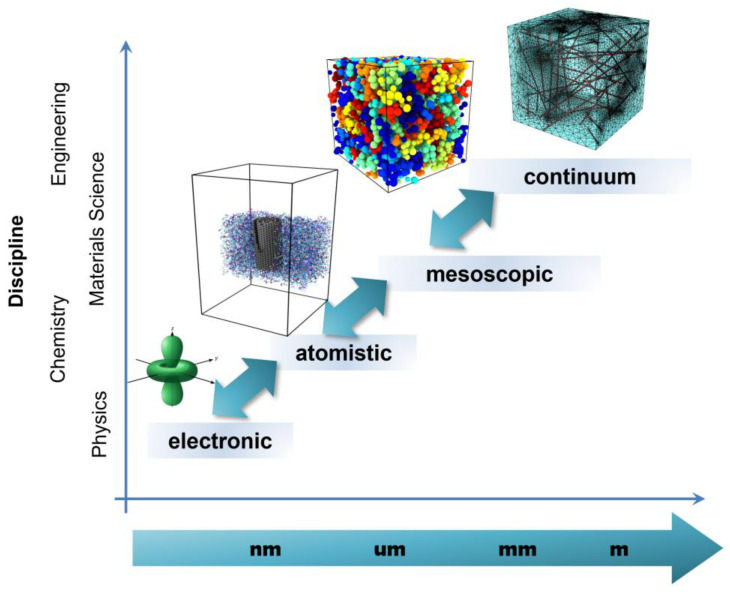
Multiscale materials modeling approach: discipline vs. length scale.

**Figure 2 sensors-24-07665-f002:**
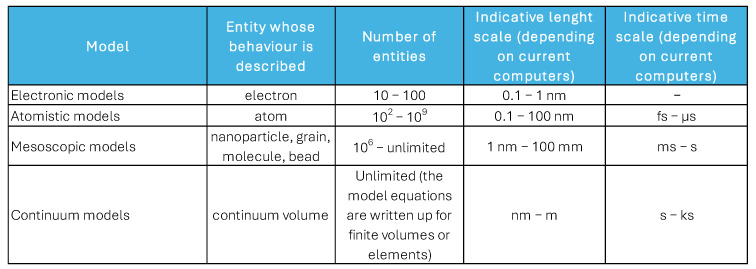
Material models, entity, length scale, and time scale [1].

**Figure 3 sensors-24-07665-f003:**
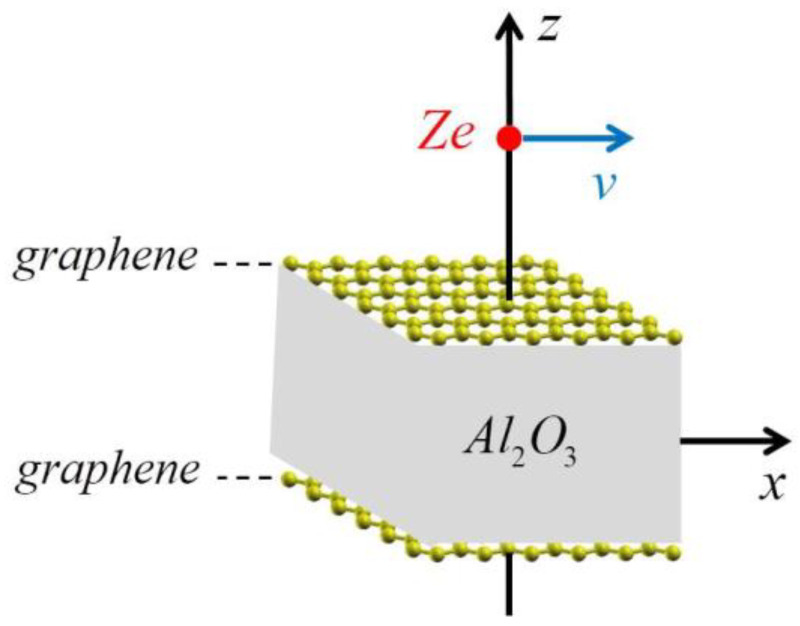
Material diagram of the graphene-Al_2_O_3_-graphene heterostructure with point charge *Ze* moving with constant speed *v* at a fixed distance above the top graphene.

**Figure 6 sensors-24-07665-f006:**
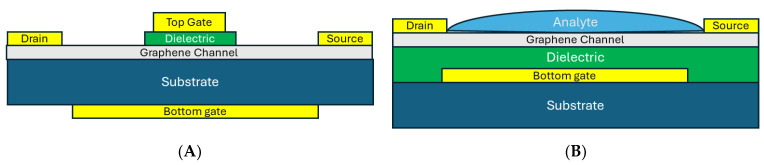
G-FET in a typical dual-gate configuration (**A**) and for biosensing applications with the analyte in contact with the graphene layer (**B**). The application of a voltage at the liquid analyte permits a state-of-the-art dual liquid gate configuration.

**Table 1 sensors-24-07665-t001:** Research studies on multiscale models of the properties of CNM-based composite materials. Abbreviations: EMT—effective medium theory; MD—molecular dynamics; CGMD—coarse-grained MD; SPH—smoothed particle hydrodynamics; DPD—dissipative particle dynamics; FEM—finite element model; ML—machine learning; CNT—carbon nanotube; Gr—graphene; GNP—graphene nanoplatelet; GO—graphene oxide; rGO—reduced GO; PVC—polyvinyl chloride; PP—polypropylene; PDMS—polydimethylsiloxane.

Study	CNM	Matrix	Analyzed Properties	Models	Ref.
*Mechanical properties*					
Venkatesan et al. (2022)	CNT	Epoxy	Elastic modulus, tensile strength, failure strain, damage index	CGMD + FEM	[182]
Caliskan and Gulsen (2023)	GNP	Epoxy	Elastic modulus, tensile strength	MD + FEM	[183]
Ekeowa and Muthu (2024)	Gr functionalized	Epoxy	Young’s modulus, Poisson ratio, tensile strength, interphase properties	MD + FEM	[184]
Ghasemi and Yazdani (2025)	CNT	PVC	Young’s modulus, Poisson ratio, shear modulus	MD + CGMD + ML	[185]
*Thermal properties*					
Wang et al. (2021)	Gr	Epoxy	Thermal conductivity, Kapitza resistance	MD + EMT	[186]
Yang et al. (2022)	CNT, Gr	Epoxy	Thermal conductivity	SPH + DPD	[187]
Muhammad et al. (2023)	Gr	Epoxy	Thermal conductivity, specific heat capacity, glass transition temperature, elastic moduli	MD + CGMD + FEM	[149]
Muhammad et al. (2023)	Gr, GO, rGO	PP	Thermal conductivity, specific heat capacity, glass transition temperature, elastic moduli	CGMD + FEM + EMT	[112]
*Electromagnetic/sensing properties*					
Grabowski et al. (2017)	CNT	Epoxy	Electrical resistivity, Young’s modulus, Poisson ratio	MD + FEM (micro and macro)	[188]
Talamadupula and Seidel (2021)	CNT	Epoxy	Piezoresistive coefficients, electrical conductivity, elastic moduli	Tunneling model + FEM	[189]
Wu et al. (2022)	Fe/Cu particles on CNT	PDMS + Sodium alginate	Electromagnetic interference shielding effectiveness, electrical conductivity, and dielectric and magnetic losses	EMT + percolation network + propagation matrix	[190]
Liu et al. (2022)	CNT, GNP	Epoxy	Piezoresistive coefficients, electrical conductivity	Bethe lattice method, excluded volume theory	[191]
Talamadupula and Seidel (2022)	CNT	Epoxy	Piezoresistive coefficients, electrical conductivity, elastic moduli	Tunneling model + FEM	[192]

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
