# Peer review of "Modeling Carbon-Based Nanomaterials (CNMs) and Derived Composites and Devices"

_sensors, 2024, doi:10.3390/s24237665_

Round 1
Reviewer 1 Report
Comments and Suggestions for Authors
This review provides an overview of different modelling strategies and methodologies used in the field of carbon-based nanoparticles, carbon-based nanomaterials and devices. The author does list more than 100 references in the manuscript, but it only stays on the surface of the pile, lack of depth integration and analysis. Further publication is not recommended. To improve the quality and academic value of this review, more strengthened integration, insightful analysis, highlight innovation and direction in this field is necessary. Please pay attention to the following questions:
1. What is the innovation and practicality of this review?
2. About writing style, "XXX were studied in Ref.[18] ", "XXX was performed in Ref.[19]", "XXX is presented in Refs.[54-57] "……Such expressions are used many times in the article, which is just listing the references without insightful analysis.
3. It is suggested to add a future prospects section for different modelling techniques of carbon-based.
4. About Figures, please provide higher definition pictures to ensure that readers can clearly see the key data and structural features, which is elementary to effectively understand the cited works. Moreover, it is obvious that there are two Figure 5 in the manuscripte.
5. There is no table in this manuscript. It is advised to incorporate a table addressing some of the recent research articles related with different modelling strategies and methodologies about carbon-based nanomaterials and derived composites and devices.
Author Response
- This review provides an overview of different modelling strategies and methodologies used in the field of carbon-based nanoparticles, carbon-based nanomaterials and devices. The author does list more than 100 references in the manuscript, but it only stays on the surface of the pile, lack of depth integration and analysis. Further publication is not recommended. To improve the quality and academic value of this review, more strengthened integration, insightful analysis, highlight innovation and direction in this field is necessary. Please pay attention to the following questions:
We thank the Reviewer for his/her feedback. The revised version of the review has been improved to include further analyses, insights, and references, also providing some examples of multi-scale models available in the literature.
- What is the innovation and practicality of this review?
Being a review article, we are not really introducing an innovation. The objective is to summarize, analyze, and synthesize existing modelling approaches to provide a comprehensive understanding of the different methodologies and tools available, and to identify challenges, opportunities, limitations, gaps or future research directions. In this sense, the utility lies in having a quite complete and updated overview of the different types of models that can be used in this field, identifying trends and patterns in research that can guide future studies and developments, avoid duplication efforts or in some cases detect replication opportunities.
Considering the comment from the Reviewer, we have added and edited some paragraphs/parts trying to improve the review in terms of integration and depth of the analyses done.
- About writing style, "XXX were studied in Ref.[18] ", "XXX was performed in Ref.[19]", "XXX is presented in Refs.[54-57] "……Such expressions are used many times in the article, which is just listing the references without insightful analysis.
We thank the Reviewer for highlighting this. Whenever possible, we have improved the explanation and analysis of the articles considered in this review. This has been done, for example, to the following references: Ref. [18] (now Ref. [23]); Ref. [19] (now Ref. [24]); Refs. [54-57] (now Refs. [59-62]); Refs. [58-61] (now Refs. [63-66]).
- It is suggested to add a future prospects section for different modelling techniques of carbon-based.
Answering the Reviewer’s comment, prospectives have been incorporated in each section and in the conclusions.
- About Figures, please provide higher definition pictures to ensure that readers can clearly see the key data and structural features, which is elementary to effectively understand the cited works. Moreover, it is obvious that there are two Figure 5 in the manuscript.
Pictures were edited for a higher definition. The figure caption for Figure 5 was corrected.
- There is no table in this manuscript. It is advised to incorporate a table addressing some of the recent research articles related with different modelling strategies and methodologies about carbon-based nanomaterials and derived composites and devices.
For the sake of clarity, the revised manuscript includes a table listing some successful, recent case studies of multiscale models predicting the properties of CNMs.
Reviewer 2 Report
Comments and Suggestions for Authors
The manuscript provides a comprehensive review of modelling techniques, applicable to carbon-based nanomaterials as well as to composites and devices and composites based on such nanomaterials. The authors draw our attention to what they call the multiscale nature of these types of materials and systems, which require different approaches for different cases. The Reviewer does not approve this particular terminology "multiscale nature" although admits that it can be a subject of personal preferrence of a scientist. The paper cites 166 references and provides the audience with impressive and helpful reading on the subject.
Author Response
- The manuscript provides a comprehensive review of modelling techniques, applicable to carbon-based nanomaterials as well as to composites and devices and composites based on such nanomaterials. The authors draw our attention to what they call the multiscale nature of these types of materials and systems, which require different approaches for different cases. The Reviewer does not approve this particular terminology "multiscale nature" although admits that it can be a subject of personal preference of a scientist.
Thank you for your feedback. To address the preference against "multiscale nature," we have rephrased the anstract to avoid this term. The revised version now reads:
“The article emphasizes that the overall performance of these materials depends on mechanisms that operate across different time and spatial scales, requiring tailored approaches based on the material type, size, internal structure/configuration, and the specific properties of interest.”
This change preserves the intended meaning while respecting the suggested terminology adjustment.
- The paper cites 166 references and provides the audience with impressive and helpful reading on the subject.
We thank the Reviewer for the positive feedback. The revised version of the manuscript includes further references, leading to a total count of 192 discussed articles on the reviewed topic.
Reviewer 3 Report
Comments and Suggestions for Authors
1.The abstract needs to be revised. The abstract should be clear and concise. The authors focus on describing the work carried out in this paper and are missing the conclusions of the study.
2. Figure 1 needs to be adjusted. The four models correspond to four pictures, and it is recommended that redundant pictures be removed in favor of readability.
3. The third paragraph of the introduction is too long and complex. It is suggested that a brief summary of the research in each chapter be provided.
4. It is suggested that the last paragraph of the introduction be deleted, and that there is no need to reflect funding information in the body of the text.
5. Please provide references for lines 107-108 of the text.
6. Please provide references [17] [18] [49] [50] important results obtained in the study.
7. The author mentions the technique of electron energy loss spectroscopy and suggests that pictures of the relevant graphical analysis be added to the text.
8. Figures 4 and 5 are not clear. It is recommended that the figures be split and placed in the text under the relevant elements. In addition, it is recommended that more images be added to the text.
9. Please provide references for lines 332-333 of the text.
10. Please provide references for lines 336-337 of the argument in the text.
11. Some paragraphs in the text are too long and complex to understand. It is suggested that the author insert a table summarizing the application of the models mentioned in the text, as well as their advantages and disadvantages.
12. The conclusion needs to be revised for excessive repetitive language with the preceding text. The conclusion should summarize the findings discussed in the text and should not repeat the findings of the previous text. In addition, this study lacks limitations and deficiencies.
Comments on the Quality of English LanguageThe English could be improved to more clearly express the research.
Author Response
- The abstract needs to be revised. The abstract should be clear and concise. The authors focus on describing the work carried out in this paper and are missing the conclusions of the study.
The abstract has been revised considering the comment from the Reviewer.
- Figure 1 needs to be adjusted. The four models correspond to four pictures, and it is recommended that redundant pictures be removed in favor of readability.
Figure 1 has been edited according to the Reviewer’s comment.
- The third paragraph of the introduction is too long and complex. It is suggested that a brief summary of the research in each chapter be provided.
Following the Reviewer’s suggestion, this paragraph was reduced, moving some parts to the other chapters.
- It is suggested that the last paragraph of the introduction be deleted, and that there is no need to reflect funding information in the body of the text.
Following the Reviewer's suggestion, this paragraph has been removed.
- Please provide references for lines 107-108 of the text.
According to this suggestion, we have added five new references labeled as [2], [3], [4], [5], and [6].
- Please provide references [17] [18] [49] [50] important results obtained in the study.
Some new text was added in Section 2 to address this comment.
- The author mentions the technique of electron energy loss spectroscopy and suggests that pictures of the relevant graphical analysis be added to the text.
We think that including additional pictures of the relevant graphical analysis in the paper would only overburden this Review article, without bringing any essential new information. Electron energy loss spectroscopy (EELS) is a well-known experimental technique. An interested reader can easily find such pictures in the literature. We decided to add only new text on theoretical modeling of experimental EELS data.
- Figures 4 and 5 are not clear. It is recommended that the figures be split and placed in the text under the relevant elements. In addition, it is recommended that more images be added to the text.
We combined multiple images in Figures 4 and 5 to illustrate the types of models discussed in the corresponding sections (e.g., RVE, size, constituent descriptions). This approach allows us to provide a comprehensive visual overview and is commonly used in review articles to consolidate related concepts (e.g, https://doi.org/10.1016/j.est.2024.114138, https://doi.org/10.1016/j.compositesb.2021.108769, https://doi.org/10.1016/j.apmt.2024.102459).
- Please provide references for lines 332-333 of the text.
Following the Reviewer's suggestion, further references have been added.
- Please provide references for lines 336-337 of the argument in the text.
Following the Reviewer's suggestion, further references have been added.
Round 2
Reviewer 1 Report
Comments and Suggestions for Authors
Reject
Reviewer 3 Report
Comments and Suggestions for Authors
Accept in present form